# The Psychological and Wellbeing Impacts of Quarantine on Frontline Workers during COVID-19 and Beyond

**DOI:** 10.3390/ijerph20105853

**Published:** 2023-05-17

**Authors:** Oliver S. Holmes, Steven Ellen, Natasha Smallwood, Karen Willis, Clare Delaney, Leon J. Worth, Shelley Dolan, Lisa Dunlop, Geraldine McDonald, Leila Karimi, Megan Rees, Maria Ftanou

**Affiliations:** 1Psychosocial Oncology Program, Peter MacCallum Cancer Centre Melbourne, Melbourne 3000, Australia; 2Chronic Respiratory Disease Laboratory, Central Clinical School, Monash University, Melbourne 3800, Australia; 3Institute for Sport and Health, Victoria University, Melbourne 3011, Australia; 4Department of Medical Education, University of Melbourne, Melbourne 3010, Australia; 5National Centre for Infections in Cancer, Sir Peter MacCallum Department of Oncology, University of Melbourne, Melbourne 3010, Australia; 6Executive Team, Peter MacCallum Cancer Centre Melbourne, Melbourne 3000, Australia; 7Clinical Governance and Strategic Projects, Peter MacCallum Cancer Centre Melbourne, Melbourne 3000, Australia; 8Prevention and Wellbeing, Peter MacCallum Cancer Centre Melbourne, Melbourne 3000, Australia; 9Department of Psychology, School of Applied Health, RMIT University, Melbourne 3000, Australia; 10Respiratory, Sleep, and General Medicine, Royal Melbourne Hospital, Melbourne 3050, Australia; 11School of Population and Global Health, University of Melbourne, Melbourne 3010, Australia

**Keywords:** COVID-19, coronavirus, quarantine, mental health, wellbeing

## Abstract

Objective: The current study investigated the experiences, wellbeing impacts, and coping strategies of frontline workers who participated in “Hotels for Heroes”, an Australian voluntary hotel quarantine program during the COVID-19 pandemic. The program was open to those who were COVID-19 positive or exposed to COVID-19 as part of their profession. Methods: Frontline workers who had stayed in voluntary quarantine between April 2020 and March 2021 were invited to participate in a voluntary, anonymous, cross-sectional online survey including both quantitative and qualitative responses. Complete responses were collected from 106 participants, which included data on sociodemographic and occupational characteristics, experiences of the Hotels for Heroes program, and validated mental health measures. Results: Mental health problems were prevalent amongst frontline workers (e.g., moderate anxiety symptoms, severe depression symptoms, and greater than usual impact of fatigue). For some, quarantine appeared to be helpful for anxiety and burnout, but quarantine also appeared to impact anxiety, depression, and PTSD negatively, and longer stays in quarantine were associated with significantly higher coronavirus anxiety and fatigue impacts. The most widely received support in quarantine was from designated program staff; however, this was reportedly accessed by less than half of the participants. Conclusions: The current study points to specific aspects of mental health care that can be applied to participants of similar voluntary quarantine programs in the future. It seems necessary to screen for psychological needs at various stages of quarantine, and to allocate appropriate care and improve its accessibility, as many participants did not utilise the routine support offered. Support should especially target disease-related anxiety, symptoms of depression and trauma, and the impacts of fatigue. Future research is needed to clarify specific phases of need throughout quarantine programs, and the barriers for participants receiving mental health supports in these contexts.

## 1. Introduction

In December 2019, the coronavirus (COVID-19) disease was identified in Wuhan city, Hubei Province in China [1]. Since then, it has spread globally, and in March 2020 COVID-19 was declared by the World Health Organization to be a pandemic, and as of February 2023 more than 500 million confirmed cases and over six million deaths have been recorded [2]. In Australia, as of July 2022, there have been over eleven million confirmed cases and over seventeen thousand deaths [2]. In March 2020, the Australian State of Victoria entered a State of Emergency, with strict restrictions applied to the distances, times, and reasons that citizens could leave their residences. Since then, there have been at least five more major metropolitan lockdowns and several further lockdowns in various regional locations due to escalating COVID-19 cases and community transmissions [3].

Across Victoria, frontline workers include clinical and non-clinical hospital workers, paramedics, patient transport workers, patient service attendants, police officers, corrections officers, aged care workers and emergency response workers. Even before the pandemic, these professionals were identified as a population especially vulnerable to burnout, depression, anxiety, and suicide [4,5,6,7,8,9]. COVID-19 added unprecedented stressors such as increased workload, new workplace information and practices, changes to working conditions and job security, reductions in household income, PPE demands, limitations on care for patients and associated moral distress, restrictions on family visitations, and the personal risk of infection and associated stigmas [10,11,12,13]. During the pandemic, mental illnesses were significantly higher among healthcare workers compared with the general public, despite those healthcare workers having generally high resilience scores [11,14,15]. Managing the mental health and psychosocial impacts of frontline workers is not only crucial for their direct wellbeing but also for the maintenance of the wider health services that depend on the capacity and engagement of these workers for patient care [14,16,17].

There are services in Australia to support the mental health of frontline workers, such as Medicare subsidised services (medicine, psychiatry, psychology, social work, and nursing), and discipline-specific referral services (e.g., Doctor’s Health Service and Nursing and Midwifery Health Program). However, generally, the uptake of formal mental health services by frontline health workers is low due to stigma and discrimination, professional repercussions, time constraints, and the perceived low severity of their condition [18,19]. In addition, accessibility was impacted during the pandemic due to long wait times for Medicare subsidised services.

The Hotel for Heroes program was a Victorian Government initiative that provided funded accommodation in hotels or apartments for frontline workers across Victoria who had exposure to coronavirus or had a positive coronavirus diagnosis and could not safely self-isolate at home. Frontline workers voluntarily accessed this scheme between April 2020 and March 2021. While the Hotel for Heroes program was designed to reduce burden on frontline staff and their families, there may be negative psychological impacts. For example, historically it has been found that quarantine is a primary predictor of acute distress disorder for frontline workers in the short term [20], and a significant predictor of post-traumatic stress symptoms in the long term [21]. Further psychological impacts related to quarantine are exhaustion, social detachment, fear and anxiety, depression symptoms, irritability and anger, insomnia, deteriorated concentration and work performance, and reluctance to work [20,21,22,23,24]. Many healthcare staff have reported altered behaviours post-quarantine, such as alcohol abuse [25], and avoidance of work and patients [24].

Given past negative impacts of voluntary quarantine programs, it is important to examine Hotels for Heroes in order to understand user experiences and the potential mental health impacts of the program. Our study aimed to explore and understand the mental health and psychosocial needs of frontline workers who utilised the voluntary Hotels for Heroes quarantine program during the COVID-19 pandemic. This included strategies used to stay psychologically well, barriers to accessing mental health supports, gaps in service delivery, and whether any additional supports were required to re-enter the workplace or meet the challenges of the future. This information is crucial for informing future services to support frontline workers to minimise the psychosocial risks of quarantine and maintain frontline workers’ capacity and engagement in providing care for patients during and beyond the COVID-19 pandemic.

## 2. Materials and Methods

### 2.1. Study Design and Recruitment

Using administrative records from the government’s Hotel for Heroes initiative in Victoria, Australia, an email was sent to 947 frontline workers who voluntarily took part in the quarantine program between April 2020 and March 2021. This included clinical and non-clinical hospital staff, paramedics, patient transport staff, patient service attendants, police officers, corrections officers, aged care workers and emergency response workers. All participants received two reminder emails over four weeks after receiving their original invitation to participate. Data were collected between February and June 2021, with each participant providing their data at one time point. The study was approved by the Peter MacCallum Cancer Centre Human Research Ethics Committee (reference number: EC00235).

The online survey was hosted on REDcap and consisted of 188 items taking approximately 15–25 min to complete. It was intended that the survey be able to capture both positive and negative aspects of the Hotels for Heroes quarantine, in order to test for the intended benefits of the program for frontline staff, while also being able to highlight areas for improvement in future programs of a similar kind. Most items were modelled based on the survey instrument used by Smallwood et al.’s (2021) investigation into frontline workers, and focused on domains including sociodemographic information, domestic and caring responsibilities, occupational characteristics, and history of physical and mental health. Thirty-eight items asked about experiences of the COVID-19 pandemic and the Hotels for Heroes program, including questions about COVID-19 training and exposure, testing incidents, confirmed COVID-19 diagnosis, circumstances and reasons for entering quarantine, length of time in quarantine, conditions on leaving quarantine, impact of quarantine on work and income, impact of quarantine on relationships and mental health, and coping strategies used during quarantine. The survey included open-ended free-text questions that asked about strengths and challenges of the program and any strategies and tools that could help manage the emotional challenges associated with quarantine.

The survey also included seven validated mental health measures: the Connor–Davidson Resilience Scale (CD-RISC-2) [26] for resilience; the Generalised Anxiety Disorder (GAD-7) scale [27] for anxiety symptoms; the Patient Health Questionnaire (PHQ-9) [28] for depression symptoms; the Impact of Life Events Scale (IES-6) [29] for psychological impact of traumatic events; the Functional Assessment of Chronic Illness Therapy scale (FACIT Version 4) [30] for impact of fatigue on daily life; the Coronavirus Anxiety Scale (CAS) [31] for COVID-19-related anxiety; and the Coronavirus Reassurance Seeking Scale (CRS) [32] for frequency of reassurance-seeking behaviours related to COVID-19.

### 2.2. Data Analysis

Quantitative data were analysed using IBM SPSS Statistics 22(IBM Australia Ltd, Sydney, Australia). Frequencies were calculated to analyse the sociodemographic and mental health characteristics of the participants, and their reasons for and experiences in using the Hotels for Heroes program. Independent-samples t-tests were used to test the differences in mental health measures as a function of time spent in quarantine (categorised as either fewer than or more than 14 days, as this was expected length of isolation for COVID-19 cases at the time the survey was conducted). *p* < 0.05 was taken to indicate statistical significance.

Free-text responses were imported into Excel, where researcher Holmes used an inductive approach to organise the qualitative statements into categories of responses. Rather than present a broader content analysis, which was beyond the scope of the current paper, we have presented frequencies for responses in each category in order to indicate those categories that might highlight areas of focus during development of similar initiatives.

## 3. Results

Of 170 survey responses, 106 (62%) provided complete data and were included in the analysis.

### 3.1. Participant Characteristics

As illustrated in Table 1 below, participants had a variety of professional backgrounds, primarily nursing (57, 54%) and aged care (20, 19%). Most participants were female (80, 76%), and younger than 40 years (74, 70%). Most reported their health was good (66, 62%), with most having no conditions affecting their physical health (87, 82%) or mental health (69, 65%) prior to the pandemic.

### 3.2. Experiences of Hotels for Heroes

The number of COVID-19 tests participants had undergone since the pandemic started ranged from 0 to 50 (*N* = 106, *M* = 7.14, *SD* = 7.08), with 57% reporting having had a confirmed COVID-19 diagnosis. Table 2 shows frequencies for circumstances and impacts of the Hotel for Heroes quarantine. The most common reason for entering quarantine was a positive COVID-19 test (56, 53%), though many also decided to quarantine due to a close contact (42, 40%), and some for other reasons (11, 10%), such as concern for transmitting to family/friends. Days spent in Hotels for Heroes ranged from 2 to 80 (*n* = 102, *M* = 14.36, *SD* = 8.92). In addition, 43% of the participants reported that they also had a period of quarantine at home due to exposure/infection to COVID-19.

### 3.3. Mental Health and Coping with Quarantine

One-third of the participants reported having a mental health condition diagnosed prior to the COVID-19 pandemic (34, 32%). Since the COVID-19 pandemic started, many of the participants reported experiencing anxiety (61, 58%), burnout (72, 68%), depression (37, 35%), PTSD (25, 24%), and other mental health problems (4, 4%).

Table 3 below shows the impacts of quarantine on relationships and mental health, as well as supports received during the program. A quarter of the participants indicated that the Hotels for Heroes program contributed negatively to anxiety (26, 25%), burnout (3, 3%), depression (22, 21%), PTSD (13, 12%), and other mental health problems (3, 3%). Others reported that the Hotels for Heroes program improved anxiety (27, 26%), burnout (16, 15%), depression, (4, 4%), PTSD (2, 2%), and other mental health problems (1, 1%). A majority of the participants reported that Hotels for Heroes quarantine had no effect on relationships (58, 55%), and where changes were reported they were more commonly closer or stronger, rather than worse, due to quarantine.

The most frequently reported psychological supports received during quarantine were checks from wellbeing officers in the Hotels for Heroes program (49, 46%) and general practitioner consultations (29, 27%). Most of the participants did not use a wellbeing app during quarantine (100, 94%), and a few reported that an app was either helpful (11, 10%) or not (15, 14%).

Table 4 below shows means, standard deviations, and ranges for scores on validated mental health measures. Mean scores indicated high levels of resilience, moderate anxiety symptoms, moderately severe depression symptoms, and an impact of fatigue greater than general population norms. Mean scores indicated that the impact of trauma was not acute, coronavirus-related anxiety was not dysfunctional, and coronavirus reassurance-seeking activities were within normal range.

Table 5 shows differences in mental health measures as a function of time spent in quarantine. Participants who spent more than 14 days in quarantine had significantly higher levels of coronavirus-related anxiety and impact of fatigue than those spending under 14 days in quarantine.

### 3.4. Free Text Responses Regarding Quarantine

When asked about the challenges of the Hotels for Heroes program, 47% of the participants reported feelings of loneliness and isolation (e.g., “having no physical contact with another person...”), 17% commented on unsatisfying food and nutrition (e.g., “food was often cold… didn’t get much fresh fruit/veg.”), and 14% reported having unmet medical concerns (e.g., “having to monitor my own symptoms because no-one was allowed in the room”). Other less commonly reported issues (reported by under 10% of the participants) included boredom, lack of physical activity, poor/mixed communication between support services, impact on family, difficulty adjusting, poor internet/phone connections, nervousness of leaving quarantine, distance of quarantine location from home, financial impacts, pressure to continue working in quarantine, fear of having spread COVID-19 to others before entering quarantine, fear of stigma, and lack of security at quarantine location.

In providing responses about the positive aspects of the Hotels for Heroes program, 44% of the participants mentioned feeling supported by staff (e.g., “lovely staff who communicated via emails, phone calls, and left typed messages and activities for us”), 37% indicated protection of family and friends (e.g., “it gave me peace of mind to be quarantined away from my partner who was negative. My biggest fear was passing it on”), 27% commented positively on the quarantine environment, and 23% commented positively on food provisions. Other less frequently reported benefits (reported by under 10% of the participants) included time spent on enjoyable activities and leisure, the location of quarantine, personal safety, ability to work in quarantine, and having free access to quarantine.

In responding to what would help in dealing with stress, anxieties, and other mental health issues related to the Hotels for Heroes program, 18% of the participants responded with mental health support (e.g., “phone calls from a psychologist employed by the Department of Health and Human Services specifically for this situation”), 16% reported social supports (e.g., “support of friends and regular wellbeing checks”), and 11% reported a need for activities (e.g., “having activities sent to the room for stimulation and something to do”). Other recommendations (reported by under 10% of the participants) included access to outside spaces, access to exercise, better food, improved communication, medical tests/care, leisure time, better quarantine location, job security, continued work, support in leaving quarantine, and knowing about future availability of quarantine.

## 4. Discussion

The current study aimed to explore and understand the mental health and psychosocial needs of frontline workers who entered quarantine during COVID-19, including strategies used to stay psychologically well, barriers to accessing mental health support, gaps in service delivery, and whether any additional supports were required to re-enter the workplace or meet the challenges of the future. Results from this study can be compared with large-scale Australian studies into the impacts of the COVID-19 pandemic on frontline workers more broadly.

Results were in line with previous findings, indicating that frontline workers faced greater mental health impacts of the COVID-19 pandemic than the general public, despite having higher levels of resilience [11,24]. However, findings regarding the mental health impact of Hotels for Heroes quarantine were mixed. A moderate portion of the participants reported that quarantine was helpful for their anxiety and burnout, and qualitative results indicated that some participants felt supported by staff, appreciated the protection for their loved ones, and were happy with the provision of comfortable food and lodgings. Impacts of the quarantine program on relationships seemed to be either negligible or trending towards having positive influences. However, many participants reported that quarantine had a negative impact on their anxiety, as well as experiences of depression and PTSD. These negative impacts are in line with previous research into the mental health impacts of quarantines [19,20], as are the qualitative reports that participants felt challenged by isolation and a need for more professional and social supports in quarantine. For workers who spent greater amounts of time in quarantine (i.e., ≥14 days), coronavirus-related anxiety and fatigue impacts appeared to be significantly more severe. The most common forms of support accessed were wellbeing checks by designated staff, but these passive forms of support were still reportedly accessed by less than half of the participants despite many reporting that professional psychological support tailored specifically to the quarantine situation would have been helpful.

Although frontline workers were shown to be resilient, there is room to increase engagement in support services as they navigate quarantine programs. A stepped-care approach may improve the capacity for services to allocate professional psychology support to those who need it, especially if screening is done at various stages (e.g., on entering quarantine, half-way through quarantine, and at exit of quarantine). For frontline workers who demonstrate need, psychological first aid could be offered as a way to reduce immediate distress and assist with the development of adaptive short- and long-term coping strategies. Psychological first aid focuses on increasing a sense of safety, calm, self- and community-efficacy, connectedness, and hope [33]. This approach seems likely to complement quarantine programs, as psychological first aid was designed to be implemented across various scenarios that risk traumatisation, and has an emphasis on the supports and resources needed to return to normalcy.

## 5. Limitations and Further Research

Of the frontline workers who used the Hotel for Heroes program, the factors that impacted participation in the current study are unknown, and so there may be selection biases that have not been addressed. Furthermore, the current cross-sectional data are subject to participants recalling experiences of the Hotel for Heroes program and its effects, and do not include comparison tests between frontline workers who did and did not participate in the quarantine program. Therefore, it may be difficult to distinguish impacts of the Hotels for Heroes quarantine program from population effects of the pandemic.

Participants reported that the Hotels for Heroes program could have been improved by greater levels of social support and professional mental health support; however, the prevalence of the participants even passively accessing support from designated support staff was less than half, and the current dataset is unable to explore this sort of discrepancy in more detail. More in-depth qualitative research is needed to explore and understand the barriers facing frontline workers in accessing and benefiting from support in the context of their quarantine experiences. Such research could also investigate in more detail the relationships between participant characteristics, personal circumstances, and their experiences of the voluntary quarantine program.

## 6. Conclusions

The COVID-19 pandemic has exacerbated mental health risks for the already vulnerable population of frontline health workers. The Hotel for Heroes program, which provided funded accommodation, seemed to have the potential to improve symptoms of anxiety and burnout, and negatively impact symptoms of anxiety, depression, and trauma. However, coronavirus-related anxiety and fatigue appeared to be significantly greater for participants who spent prolonged periods of time in quarantine. These findings point to specific aspects of mental health care that can be applied to participants of similar voluntary quarantine programs in the future. It seems necessary to screen for psychological needs at various stages of quarantine, and to allocate appropriate care and improve its accessibility, as many participants did not utilise the routine support offered. Support should especially target disease-related anxiety, symptoms of depression and trauma, and the impacts of fatigue. Further qualitative research would assist in understanding how frontline workers using quarantine can be best offered support to target these symptoms, and to overcome barriers of seeking support, so that the benefits of quarantine programs can be achieved while minimising their negative impacts.

## Figures and Tables

**Table 1 ijerph-20-05853-t001:** Participant characteristics (*N* = 106).

Characteristic	Frequency	Percent
**Age (years)**		
18–30	45	42.5
31–40	29	27.4
41–50	14	13.2
>50	18	17
**Gender**		
Male	25	23.6
Female	80	75.5
Non-binary	1	0.9
**No. of people in household**		
Lives alone (1 person)	9	8.5
Lives with 1 or more others	97	91.5
**No. of children <16 years/older adults at home**		
1–2	26	24.5
3+	1	0.9
Lives with ≥1 elderly person/people at home	15	14.2
**Need to manage home schooling during pandemic**		
Yes	23	21.7
No	44	41.5
Not applicable	39	36.8
**Caring duties during pandemic**		
Yes	29	27.4
No	60	56.6
Not applicable	17	16
**Professional background**		
Nursing	57	53.8
Aged care worker	20	18.9
Medical	6	5.7
Allied health	10	9.4
Administrative staff	1	0.9
Police	2	1.9
Paramedic	3	2.8
Other roles	7	6.6
**Years worked in profession**		
0–5 years	52	49.1
6–10 years	21	19.8
11–15 years	14	13.2
15+ years	19	17.9
**Frontline area worked during pandemic**		
Aged care	28	26.4
Community care	1	0.9
Hospital setting	66	62.3
Primary care	2	1.9
Police	2	1.9
Paramedic	3	2.8
Disability care/outreach	4	3.8
**Physical health status**		
Excellent	27	25.5
Good	66	62.3
Fair	10	9.4
Poor	3	2.8
**Health condition prior to the pandemic**		
Yes	19	17.9
No	87	82.1

**Table 2 ijerph-20-05853-t002:** Impacts of Hotels for Heroes Quarantine (*N* = 106).

	Frequency	Percent
**Reason for quarantine ***		
Positive COVID-19 test	56	52.8
Close contact of COVID-19 positive individual	42	39.6
Other **	11	10.4
**Employment status before quarantine**		
Full time	39	36.8
Part time	52	49.1
Casual	13	12.3
Other	2	1.9
**Current employment status**		
Full time	41	38.7
Part time	50	47.2
Casual	12	11.3
Other	3	2.8
**Change in work since quarantine ***		
Increased paid hours	11	10.4
Increased unpaid hours	3	2.8
Decreased hours (paid or unpaid)	12	11.3
No change	81	76.4
**Redeployment since quarantine**		
Yes	19	17.9
No	87	82.1
**Change in work role since quarantine**		
Yes	24	22.6
No	82	77.4
**Impact of quarantine on household income**		
Increased	2	1.9
Decreased	18	17
No change	86	81.1
**Concerns or worries about household income due to quarantine**		
Yes	16	15.1
No	90	84.9

* Multiple choices permitted. ** Includes concerns about infecting family/friends/housemates.

**Table 3 ijerph-20-05853-t003:** Impacts of quarantine on relationships, mental health, and supports (*N* = 106).

	Frequency	Percent
**Quarantine impact on relationships ***		
I have a *closer or stronger* relationship with my partner	16	15.1
I have a *worse* relationship with my partner	6	5.7
I have a *closer or stronger* relationship with my children/parents/family	26	24.5
I have a *worse* relationship with my children/parents/family	4	3.8
I have a *closer or stronger* relationship with my friends	20	18.9
I have a *worse* relationship with my friends	6	5.7
I have a *closer or stronger* relationship with my work colleagues	20	18.9
I have a *worse* relationship with my work colleagues	5	4.7
No effect on relationships	58	54.7
**Negative impacts of quarantine ***		
Anxiety	26	24.5
Burn out	3	2.8
Depression	22	20.8
PTSD	13	12.3
Other mental health problem	3	2.8
None of the above	68	64.2
Prefer not to say	2	1.9
**Positive impacts of quarantine ***		
Anxiety	27	25.5
Burn out	16	15.1
Depression	4	3.8
PTSD	2	1.9
Other mental health problem	1	0.9
None of the above	65	61.3
Prefer not to say	4	3.8
**Support received during quarantine ***		
Wellbeing seminar provided by Hotels for Heroes	6	5.7
Victorian Government wellbeing website	13	12.3
Wellbeing checks by wellbeing officer	49	46.2
General practitioner	29	27.4
Employee Assistance Program	15	14.2
Community counselling	10	9.4
None of the above	28	26.4
Other supports **	21	19.8
**Wellbeing app used during quarantine?**		
Yes ***	6	5.7
No	100	94.3
**App usefulness**		
Yes	11	10.4
No	15	14.2
N/A	4	3.8

* Multiple choices permitted. ** Includes activity program, online exercise groups, contact with family/friends/colleagues, counselling/therapy, and spiritual community. *** Includes Calm, Sanity & Self, Headspace, Insight Timer, Meditation, Smiling Mind, Treat, and Relaxation Audio.

**Table 4 ijerph-20-05853-t004:** Means, standard deviations, and ranges of mental health measures (*N* = 106).

	*M*	*SD*	*Range*
**Resilience**	8.33	1.41	5–10
**GAD-7**	13.55	5.88	7–28
**PHQ-9**	16.81	6.98	9–36
**IES-6**	7.87	6.47	0–24
**CAS**	2.31	4.11	0–20
**CRSBS**	1.61	3.73	0–20
**FACIT**	32.78	10.65	8–52

**Table 5 ijerph-20-05853-t005:** Differences in mental health measures as a function of time in Hotels for Heroes program.

	Days in Quarantine				
	≤14 Days (*n* = 79)	>14 Days (*n* = 23)	*df*	*t*	*p*
**Resilience**	8.27 (1.42)	8.57 (1.47)	100	−0.88	0.190
**GAD-7**	13.04 (5.79)	14.83 (6.00)	100	−1.29	0.099
**PHQ-9**	16.46 (6.90)	17.91 (7.69)	100	−0.87	0.193
**IES-6**	7.56 (6.35)	9.04 (6.72)	100	−0.98	0.166
**CAS**	1.76 (3.43)	4.17 (5.72)	26.77	−1.93	0.032 *
**CRSBS**	1.30 (3.06)	2.35 (5.00)	26.97	−0.95	0.175
**FACIT**	33.88 (10.67)	29.48 (10.67)	100	1.74	0.042 *

*Note.* Figures in parentheses are standard deviations. * *p* < 0.05, ** *p* < 0.01, *** *p* < 0.001.

## Data Availability

The data presented in this study are available on request from the corresponding author. The data are not publicly available as per ethical approvals.

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
