# Peer review of "The Psychological and Wellbeing Impacts of Quarantine on Frontline Workers during COVID-19 and Beyond"

_ijerph, 2023, doi:10.3390/ijerph20105853_

Round 1
Reviewer 1 Report
The purpose of the present study is to evaluate the mental state of the workers who participated in the „Hotel for Heroes“ program during the Covid 19 pandemic.
The topic of the article sounds interesting.
Abstract:
Objective: In general, the abstract is watered down. The goal is not well defined. What exactly is being studied? Both the Covid pandemic itself and the quarantine itself have affected the mental state of almost the entire population. What is the purpose of the present study in this case? A reformulation is needed.
Results: To provide information on the difference between those workers who received psychological help under a certain program and those who did not have access to it. If no such data are available, reference should be made to the limitations of this study.
The conclusion of the abstract is also not informative. Of course, the quarantine will have both positive and negative effects. Is a distinction made between when it is positive and when the effect is negative? What factors determine and influence this difference?
The introduction needs an analysis of the controversies in the literature on both the impact of Covid and the introduction of quarantine measures. Some countries did not impose a strict ones. What is the impact of them and what are the data related to it. Are there similar programs in other countries also affected by the Covid 19 pandemic.
Methods:
The participation in the quarantine was during the period from 2019 to 2021. It is not clear how much time passed between coming out of quarantine and when the study was conducted. Experiences would have dynamics over time, and it is not clear whether the stress study is current or retrospective in nature.
Statistical methods are correctly selected.
Results:
What does the answer line 182-184 mean. At "Nearly two-thirds of participants reported that Hotels for Heroes had no negative impacts on their mental health (68, 64%)“. What this means ? How do you interpret it? Rephrasing need.
It is recommended to divide the table into separate tables, because it is difficult to get a clear idea of ​​the research done. It is necessary to make a comment on the individual indicators in it.
3.4. Free Text Responses Regarding Quarantine
In this section, it is necessary to explain what is the difference in the answers that would be given by other people who do not work on the front line and did not visit the "Hotel for Heroes", but spent their quarantine in a home environment. The goal is to point out the differences and features related to the tasks.
The discussion needs to be expanded. It is necessary to point out the possible differences between the persons quarantined at home and those in the Heroes' hotel. Do you think there will be a difference between them? Are there differences between frontline workers who live alone and those who have a family / or are in a serious relationship /.
You presented data on education but not on its impact on the analyzed indicators.
A "Limitations" section needs to be presented
It should present the limitations of the present study as well as outline the directions for possible future studies if you consider that there is a need for such.
The Conclusion section should be revised accordingly.
The reviewer.
Author Response
Thank you for your thoughtful feedback for our manuscript. Your comments and suggestions have significantly improved the quality of the paper. Please find our responses attached.

Reviewer 2 Report
This study aimed to explore and understand the mental health and psychosocial needs of frontline workers who have entered quarantine during COVID-19 and whether additional support is required to reenter the workplace or meet future challenges. The findings help future services to support frontline workers to minimize the psychosocial risks of quarantine and maintain frontline workers' capacity and engagement in providing care for patients during and beyond the COVID-19 pandemic.
pp.8-9, ln. 258-262
"A moderate portion of participants reported that quarantine was helpful for their anxiety and burnout, and qualitative results indicated that some participants felt supported by staff, the protection for their loved ones, and the provision of comfortable food and lodgings. However, many participants reported that quarantine had a negative impact on their anxiety, as well as experiences of depression and PTSD."
The finding that quarantine positively impacted a moderate portion of participants and negatively impacted many other participants is interesting but leaves room for further analysis. The mental health impacts of quarantines were related to the duration of the quarantine. The authors might also clarify the relationship with the participants' attributes, such as sociodemographic information, domestic and caring responsibilities, occupational characteristics, and physical and mental health history. Please attempt to discuss why the impact of quarantine is divided between participants as positive or negative.
Author Response

(The authors gave the same response as above.)

Round 2
Reviewer 1 Report
Thank you for answering the questions.
Despite the thorough attempt to revise the article, I think the conclusion and abstract need further revision to make it clearer what "take home massage" the reader will take away.
Reviewer
Author Response
Dear reviewer,
Thank you for your continued feedback that has helped to improve the quality of our paper. Based on your suggestions we have updated the abstract and the conclusion to clarify the central take away messages from our findings.
Kind regards
